# Epigenome Modulation Induced by Ketogenic Diets

**DOI:** 10.3390/nu14153245

**Published:** 2022-08-08

**Authors:** Paola Ungaro, Immacolata Cristina Nettore, Fabiana Franchini, Giuseppe Palatucci, Giovanna Muscogiuri, Annamaria Colao, Paolo Emidio Macchia

**Affiliations:** 1Istituto per l’Endocrinologia e l’Oncologia Sperimentale (IEOS) “Gaetano Salvatore”, Consiglio Nazionale delle Ricerche, 80131 Naples, Italy; 2UNESCO Chair on Health Education and Sustainable Development, Università degli Studi di Napoli Federico II, 80131 Naples, Italy; 3Dipartimento di Medicina Clinica e Chirurgia, Scuola di Medicina, Università degli Studi di Napoli Federico II, 80131 Naples, Italy

**Keywords:** ketogenic diet, Very Low-Calories Ketogenic Diet, DNA methylation, histone modifications, miRNAs

## Abstract

Ketogenic diets (KD) are dietary strategies low in carbohydrates, normal in protein, and high, normal, or reduced in fat with or without (Very Low-Calories Ketogenic Diet, VLCKD) a reduced caloric intake. KDs have been shown to be useful in the treatment of obesity, metabolic diseases and related disorders, neurological diseases, and various pathological conditions such as cancer, nonalcoholic liver disease, and chronic pain. Several studies have investigated the intracellular metabolic pathways that contribute to the beneficial effects of these diets. Although epigenetic changes are among the most important determinants of an organism’s ability to adapt to environmental changes, data on the epigenetic changes associated with these dietary pathways are still limited. This review provides an overview of the major epigenetic changes associated with KDs.

## 1. Introduction

Ketogenic diets (KD) are a group of diets characterized by low carbohydrate, normal protein, and high fat content. This combination of macronutrients aims to force the body to break down fat instead of glucose to synthesize adenosine triphosphate, inducing ketosis [1] and mimicking the metabolic state of starvation or fasting. In general, urinary ketone levels can be used as an indicator of dietary adherence [2]. Several variations of the KD have been proposed [3]. The best studied is a KD with a 4:1 or 3:1 ratio of dietary fat to combined dietary protein and carbohydrate, sometimes referred to as a “classic” KD [3].

The earliest evidence showing the role of a ketogenic diet dates back to the 1920s, when it was discovered that by eliminating carbohydrates from the diet, the metabolic effects of fasting could be mimicked. KD results in a marked increase in circulating concentrations of β-hydroxybutyrate (β-HB) [4]. β-HB is the main representative of ketone bodies in animal cells, and its baseline concentrations in serum are in the micromolar range in humans. During fasting or after carbohydrate-free KD, serum concentrations of β-HB can rise well above 2 mM [5].

This dietary approach was first used in pediatrics to treat drug-resistant epilepsies [6]. Later, the studies of Cahill and his group were critical in clarifying the physiology of starvation and the protein-sparing effects of ketosis [7,8]. The authors demonstrated that ketones reduce muscle protein breakdown during starvation by providing alternative fuel to the brain, thereby reducing the need for gluconeogenesis. In addition, the decrease in insulin levels caused by caloric deprivation and carbohydrate restriction allows the production of glucose through glycogenolysis and gluconeogenesis to fuel the brain. In addition, free fatty acids are released from adipose tissue and used as a substrate for the liver production of ketone bodies (acetoacetate, beta-hydroxybutyrate, and acetone). These molecules gradually become an alternative source of energy for the body and brain [8,9]. Finally, the metabolic changes induced by fasting, due to the anorexic properties of ketone bodies, allow a reduction in the sensation of hunger and reduce protein breakdown in the body, allowing the organism to survive [10]. This model has been defined as “Protein Sparing Modified Fast” (PSMF) and consists of a highly hypocaloric diet, based mainly on an adequate intake of proteins and essential fatty acids, which allows rapid weight loss while maintaining muscle mass [11]. This type of treatment may even be therapeutic in certain pathological conditions [12].

The Very Low-Calorie Ketogenic Diet (VLCKD) is a nutritional intervention characterized by a very low carbohydrate content (equivalent to 5–10% of the total daily caloric intake or 30–50 g of carbohydrate per day [13,14]), a low fat content, and a high-biological-value protein content of 0.8 to 1.2 g/kg of ideal body weight [15]. VLCKD promotes metabolic changes and optimizes energy metabolism by simulating the effects of fasting, with a drastic restriction of the daily carbohydrate intake and a relative proportional increase of fats and proteins for a total caloric intake of about 800 kcal/day [16,17].

In the past century, the use of diets with very severe caloric restriction was not uncommon. In some cases, the treatment consisted in the administration of water-only diets and total fasting (with or without vitamin supplements, depending on the duration of the diet) [10]. The first report on the effects of prolonged fasting was published by Benedict in 1915 [18], followed by Keys in 1940 [19]. Both authors found that fasting resulted in weight loss in normal-weight individuals, mainly at the expense of lean mass [18,19]. In subsequent years, treatment of obesity by complete fasting was introduced [20,21]. In obese individuals, the weight loss induced by fasting was less relevant in terms of fat-free mass consumption due to increased fat reserves. Consequently, in these individuals, the duration of fasting could be extended from two months (in lean individuals) [22] to almost one year [23]. Complete [23,24] or intermittent [25,26] fasting was then more commonly used to treat obesity. Although fasting diets were generally well tolerated, they were associated with several adverse effects, including malnutrition, rapid weight loss, heart disease, hypotension, and hypoglycemia [10,27,28].

Nowadays, KD and VLCKD are considered valid therapeutic options in various clinical situations, including severe obesity, obesity in the presence of concomitant diseases, hepatic steatosis, polycystic ovary syndrome, drug-resistant epilepsy, migraine, and in the preoperative management of bariatric surgery [16,29]. In recent years, increasing scientific evidence has favored the rediscovery of KD and VLCKD, and these approaches have been proposed as attractive nutritional strategies for the treatment of obesity in individuals who have already tried unsuccessfully to lose weight with more balanced diets [16]. In addition, KD and VLCKD have been shown to have beneficial effects on body composition, the metabolic profile, and the expression of genes related to inflammation and oxidative stress [16,29,30,31].

Although KD and VLCKD are now widely used to prevent and treat various clinical conditions [1], the molecular mechanisms underlying the beneficial effects of these diets are still largely unknown. This is a very critical issue, as the characterization of these mechanisms could enable the development of potential long-term therapeutic options to extend the benefits of carbohydrate restriction after diet discontinuation [32].

The epigenome is a link between an individual’s genetic background and the environment, and it determines an organism’s ability to respond and adapt to changes in the environment [33]. Diet is among the most important epigenetic modulators [34,35], and increasing evidence suggests that epigenetic changes are associated with nutritional deficiency or supplementation [36,37] and with ketogenic diets [38,39,40,41,42].

The term epigenetics refers to heritable changes in gene expression without changes in DNA sequence. Major epigenetic mechanisms include DNA methylation, histone modifications, microRNA, and chromatin remodeling. All of these mechanisms can bidirectionally affect the state of chromatin by changing it from “closed”, which restricts the access of transcription proteins, to “open”, which allows the access of transcription factors [43] and the initiation of transcription.

In the next pages, we will examine the epigenetic changes induced by KD and VLCKD, as these may be among the mechanisms by which KD and VLKD achieve their beneficial effects.

## 2. DNA Methylation

DNA methylation is the most studied epigenetic modification. It consists in the transfer of a methyl group from the main methyl-donor, S-adenosylmethionine (SAM), to the C5 position of cytosine within CpG dinucleotides and the formation of 5-methylcytosine [44]. This process is mediated by the action of enzymes belonging to the family of DNA methyltransferases (DNMTs) [45].

Most of the available information on the effects of KD on DNA methylation comes from a rat model suffering from chronic epilepsy or from the adjuvant treatment in drug-resistant epilepsy [46]. Administration of a ketogenic, high-fat, low-carbohydrate diet to these animals attenuated seizure progression and reduced the global DNA methylation status [47]. These effects are likely related to the KD-induced increase in adenosine in the hippocampus [48,49]: increased adenosine promotes the formation of S-adenosylhomocysteine (SAH) [50], which in turn blocks DNA methyltransferase [51], reducing global DNA methylation [50,52]. Global DNA hypermethylation has been found in patients with temporal lobe epilepsy [53], and increased concentrations of adenosine in the brain observed after KD have been proposed as the main mechanisms by which this diet improves the clinical picture in patients with drug-resistant epilepsy [52,54,55].

Complex changes in DNA methylation patterns have also been found during aging [56,57] and have been associated with various age-related diseases [39]. It has been suggested that KD or exogenous ketogenic supplements [58] may promote anti-aging effects [58,59,60,61]. It is likely that these effects occur via the modulation of the DNA methylation of the same genes that respond to changes in brain adenosine levels [50,57], including *KLF14*, *ELOVL2*, *FHL2*, *OTUD7A*, *SLC12A5*, *ZYG11A*, and *CCDC102B* genes [57].

Recently, an interesting study was published examining the effects of VLCKD for the treatment of obesity on DNA methylation patterns [62]. In the study, 850,000 CpG sites were compared, using a dedicated array-based platform, between twenty-one patients with obesity after six months of VLCKD and twelve normal-weight volunteers. After weight reduction by VLCKD, differences were found in 988 CpG sites (786 unique genes) whose methylation patterns resembled those of normal-weight volunteers, suggesting the downregulation of *DNMT1*, *DNMT3a*, and *DNMT3b* gene expression [62]. Several of the genes whose methylation status changed were involved in metabolic processes, protein metabolism, and the development of muscles, organs, and skeletal system development, confirming the hypothesis of epigenetic dysregulation in the adipose tissue of subjects with obesity [63,64]. In addition, changes in the methylation status were also detected in some novel genes, including *ZNF331*, *FGFRL1*, *CBFA2T3*, *C3orf38*, *JSRP1*, and *LRFN4*, whose methylation status was specifically associated with VLCKD-induced weight loss and/or ketosis [62].

It is well known that a specific methylation profile is associated not only with obesity in various tissues [65,66,67,68,69,70,71], but also with complications of obesity such as insulin resistance [63] and cancer [72,73]. Therefore, it is possible to hypothesize that one of the mechanisms by which VLCKD contributes to the maintenance of weight loss in the treatment of obesity is related to the changes in DNA methylation that this diet may induce [31,74].

The described effects of KD/VLCKD on DNA methylation are shown in Figure 1. Data in the literature suggest that the effects on DNA methylation caused by KD or VLCKD may be various and more complex than those reported so far; therefore, additional studies are needed to clarify these interplays.

## 3. Histone Modifications

The genetic information in the nucleus of eukaryotic cells is organized in a highly conserved structural polymer called chromatin [75]. The basic unit of chromatin is the nucleosome. The nucleosome consists of 147 bp of DNA wrapped around an octamer of histones composed of two H3/H4 dimers and two H2A/H2B dimers [33]. The dynamic and reversible modifications of histone proteins influence all DNA-based processes, including DNA packaging and chromatin compaction, nucleosome dynamics, and transcription [76,77]. This process is regulated by several post-translational modifications, most of which occur within the amino terminal in histone tails [78].

Recently, ketone bodies have been shown to act as epigenetic modifiers that determine covalent modifications at key histones [4,40,79]. These include lysine acetylation, methylation, and a novel epigenetic modification, lysine β-hydroxybutyrylation (Kbhb) [80]. Since Kbhb sites overlap with lysine residues, this novel histone modification integrates the classic histone lysine acetylation, methylation, and histone phosphorylation and ubiquitination, producing changes in the chromatin status (Figure 1). Generally, a genome-wide analysis (ChIP-swq) associated with transcriptional profiling revealed that the β-hydroxybutyrylation of histones is associated with active gene transcription. In liver from mice undergoing long-term fasting or streptozotocin-induced diabetic ketoacidosis, histone lysines susceptible to beta-hydroxybutyrylation were identified, including H1K168, H2AK5/K125, H2BK20, H3K4/K9/K14/K23, and H4K8/K12 [4]. In their study, the authors demonstrate that upregulated genes bearing the H3K9ac and H3K4me3 marks are different from those carrying the H3K9bhb mark, suggesting that histone Kbhb has different transcriptional-promoting functions than histone acetylation and methylation [80,81].

The effects of D-3-hydroxybutyric acid (BHB), one of the most abundant ketone bodies, on the establishment of histone tails’ posttranslational modifications are very controversial. Initially, BHB was considered an endogenous inhibitor of class I and class II histone deacetylases (HDACs); indeed, it has been shown that BHB treatment, in HEK293 cells or high levels of BHB produced by a subcutaneous pump delivery in C57BL6/J mice maintained on caloric restriction [82,83], produced a dose-dependent histone hyperacetylation, especially on lysine 9 and 14 of histone 3 (H3K9/K14). Other studies suggest that BHB only induces very marginal changes in the acetylation patterns, i.e., experiments conducted in various cell lines such as HEK293 cells, myotubes (L6), and endothelial cells (HMEC-1) showed that BHB administration had no effect on histone acetylation and did not inhibit histone deacetylase catalytic activity [84].

Understanding the effect of BHB on chromatin acetylation patterns is complicated by the NAD+/NADH ratio, which depends significantly on the presence of BHB or glucose.

BHB can substitute glucose as fuel; however, while glucose is able to produce four moles of NADH from NAD+, BHB is able to reduce only two moles of NAD+. The excess of the NAD+ resulting from the ketogenic diet modulates the activity of NAD+-dependent enzymes such as the Sirtuin proteins (SIRT1-7), which are involved in the deacetylation process [85] (Figure 1).

Mice fed a ketogenic, low-carbohydrate diet showed a global increase in protein acetylation. These changes were accompanied by modifications in the regulation of rapamycin complex 1 (mTORC1), and in older animals, motor function and memory were preserved and tumor incidence was lower [60]. Similarly, obesity, higher mortality, and age-related memory loss were prevented in mice fed a cyclic (every two weeks) ketogenic diet [83]. These effects have been linked to the downregulation of genes involved in fatty acid metabolism and the upregulation of Peroxisome Proliferator-Activated Receptor alpha (PPAR-) target genes [83], mechanisms that have been shown to be closely linked to epigenetic changes, at least in adipose tissue [64,86].

Further evidence for the role of epigenetic regulation by KD has been obtained in several mouse models. In mice exposed to hypobaric hypoxia, KD increased histone acetylation and rescued spatial memory impairment [87]. Moreover, in Kmt2d(+/betaGeo), a mouse model with Kabuki syndrome, KD had the same effect as the administration of AR-42, a histone deacetylase inhibitor that promotes chromatin opening and rescues both the neurogenesis defect and hippocampal memory impairment. These effects are mediated via an increase in βOHB, which in turn modulates H3 acetylase and H3K4me3 [88].

## 4. MicroRNAs

MicroRNAs (miRNAs) are small non-coding RNAs consisting of about 20–22 nucleotides [89]. miRNAs are formed during the processing of longer RNA transcripts [90] and regulate gene expression by binding to target mRNAs to modulate their stability, adjust their intracellular concentrations, and control their translation into proteins [89,91]. It is estimated that only a few hundred miRNAs regulate between 30 and 80% of the genes encoded in the human genome [92]. Numerous miRNAs have been identified in human adipose tissue, and recent data suggest that there are several circulating miRNAs that are differentially expressed and can be detected in the serum of people with obesity compared with lean people. These can be used to predict the risk of obesity-related complications and to monitor the outcomes of weight loss interventions [93]. In addition, miRNAs have emerged as attractive therapeutic agents for various diseases in recent years, and miRNA-based therapeutic tools have been tested in preclinical studies and clinical trials [93]. Finally, recent preliminary findings by Caradonna and coworkers suggest that circulating miRNA levels may be modulated by specific dietary interventions [94]. Many of them, like the Mediterranean Diet, have been demonstrated to protect from chronic degenerative diseases and have also been proposed as an additional therapeutic option in several clinical conditions, suggesting possible links between miRNA, diet, and aging or age-related inflammation [94,95].

In animal models, caloric restriction has been shown to modulate circulating microRNAs. A very recent study examined miRNA levels in the serum and tissues of mice fed 40% CR for 28 days [96]. The results showed that miR-16-5p, miR-196b-5p, and miR-218-5p were increased in serum and that miR-16-5p was also increased in the spleen, thymus, colon, and stomach of CR animals. Of note, miR-16-5p is able to downregulate the expression of inflammatory cytokines, suggesting that CR can indirectly modulate inflammation [96].

To date, few human studies have investigated the effects of KD or VLCKD on miRNA. In 2019, Cannataro and coworkers examined the miRNA profile of 36 subjects with obesity who underwent six weeks of biphasic KD. The authors examined 799 circulating miRNAs and found that changes in only three of them (hsa-let-7b-5p, hsa-miR-143-3p, and hsa-miR-504-5p) were associated with KD in a sex-independent manner. The targets of these miRNAs were involved in cytokine signaling pathways, nutrient metabolism, oxidative phosphorylation, functional regulation of PPARs, and insulin signaling [97]. The same authors have also shown that miRNAs associated with antioxidant and anti-inflammatory signaling pathways are altered in individuals with obesity but return to levels comparable to those of lean individuals after KD [98].

Considering the multiple potential targets of these miRNAs, it is possible to use KD and VLCKD as an adjuvant therapeutic option in various diseases, including neurological disorders and cancer [97].

## 5. Conclusions

The data presented here suggest that KD and VLCKD achieve some of their beneficial effects via the modulation of the epigenome, resulting in changes in DNA methylation, histone modifications, and miRNA levels (Table 1). Although the data presented here are still preliminary and limited to a few reports, the relationship between ketosis and epigenetic changes is very interesting, and this area clearly needs further investigation.

A better understanding of the relationship between diet, obesity, and epigenetic changes induced by KD or VLCKD could have important consequences for elucidating the effects of ketosis, not only in terms of weight loss but also for the possible use of KD or VLCKD as supportive treatment for various diseases, as well as to reduce the side effects associated with these diets. Indeed, several reports have indicated that KD or VLCKD are associated with some adverse effects. Constipation, headache, bad breath, muscle cramps, diarrhea, vomiting, and general weakness have been reported during the first few weeks of the diet. These symptoms, termed “keto flu,” are among the factors that may limit adherence to these diets [99,100]. In addition, some important potential cardiovascular side effects of KD have also been reported [100]. These include an increase in triglycerides and low-density lipoprotein (LDL) cholesterol [101], a transient increase in endothelial dysfunction [102], an increase in cardiac insulin resistance [103], and various others side effects [99,100]. There are no data in the literature examining the molecular mechanisms by which KD and VLCKD contribute to these adverse effects, and it cannot be ruled out that these mechanisms are also mediated by the modulation of the epigenome.

It is possible that additional strategies to stabilize the epigenetic changes induced by ketosis may contribute to the long-term success of KD. Among other things, it cannot be ruled out that combining KD/VLCKD with other nutrients whose consumption has been shown to have epigenetic effects may help to enhance and/or stabilize epigenetic changes and thus support the success of nutritional programs. This could enhance and prolong the global chromatin changes induced by ketosis, thus promoting the beneficial effects associated with these diets. In conclusion, this is a new field of research that will open up interesting possibilities for the treatment and prevention of obesity, metabolic disorders, and various chronic diseases in the near future.

## Figures and Tables

**Figure 1 nutrients-14-03245-f001:**
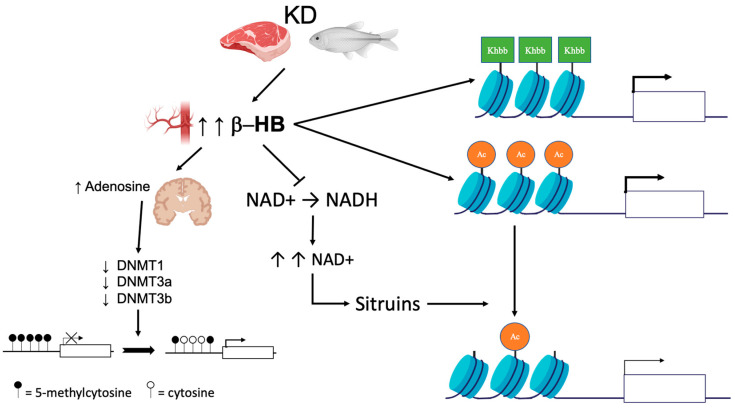
Epigenetic changes induced by KD. KD leads to an increase in circulating ketone bodies (β-hydroxybutyrate [β-HB]). In the brain, higher concentrations of β-HB have inhibitory effects on DNA methyltransferases (DNMT) via an increase in adenosine, which in turn leads to global DNA hypomethylation. Ketone bodies also act as epigenetic modifiers that determine covalent modifications at key histones such as lysine acetylation (Ac), methylation, and β-hydroxybutyrylation (Kbhb). Finally, β-HB is less efficient than glucose as a substrate for NADH production. The increase in the NAD + level promotes histone demethylation via the modulation of sirtuin activity.

**Table 1 nutrients-14-03245-t001:** Epigenetic changes determined by KD.

Epigenetic Modifications	Effect of Ketosis	Possible Mechanisms	Subject of Investigation	Refs
DNA methylation	Global DNA hypomethylation	Increased brain adenosine	Rats	[47]
			Human	[52,54,55]
		Modulation of genes regulating DNA methylation	Rats	[50]
			Human	[57]
		Downregulation of *DNMT1*, *DNMT3a*, and *DNMT3b*	Human (subjects with obesity)	[62]
Histone modifications	Covalent modifications to key histones	Lysine acetylation, methylation, and β-hydroxybutyrylation	Cell lines	[79]
			HEK293 cell, mice	[4,40,80]
		βOHB inhibits class I histone deacetylases	Cell lines	[40]
		βOHB increases histone acetylation	HEK293 cell line	[2,82]
		Sirtuins-mediated histone deacetylation	Cell lines	[40]
		Global increase in protein acetylation	Mice	[60]
		Increased the levels of histone acetylation	Rats	[87]
miRNAs	Elevation of miR-16-5p, miR-196b-5p, and miR-218-5p	Unknown. Changes in these miRNAs are determined by caloric restriction.	Mice	[96]
	Modifications in hsa-let-7b-5p, hsa-miR-143-3p, and hsa-miR-504-5p	Unknown. The target genes of the miRNAs are associated with obesity and metabolism-related pathways.	Human	[97]

## Data Availability

Not applicable.

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
