# Peer review of "Epigenome Modulation Induced by Ketogenic Diets"

_nutrients, 2022, doi:10.3390/nu14153245_

Round 1
Reviewer 1 Report
Major revisions
1. In rows 120-126 the authors reported some references regarding global DNA methylation in patients with epilepsy after KD treatment. It would be better to report the tissue /cells from which the analyses were performed.
2. Similarly, in rows 127-131 please specify the genes responding in brain adenosine levels.
3. In row 138 please report exactly what the reference 62 says. In particular the DNMTs have been studied at expression level, so, the DNMT 1, 3a, 3b are genes and must be reported in Italics and uppercase.
4. Please, similarly do for genes reported in rows 142-144.
5. The sentence in rows 145-148 is confuse and contradictory. Moreover, the relationship with cell differentiation it seems inappropriate. Please reformulate the sentence or eliminate it.
6. In the first part of section 4 (MicroRNAs) it would be necessary to report the preliminary results of Caradonna et al (Science and healthy meals….. Nutrients, 2020) regarding circulating miRNAs in comparison with specific food interventions which gave rise to the question about relationship between aging, age-related inflammation, circulating miRNAs, and diet.
Minor revision
1. Please make a double check of the references numbers. In all manuscript have been often included reference ranges with a couple of number in brackets not sequential each other.
Author Response
We would like to thank the reviewer for his suggestions. All his comments have been addressed as indicated in the following list:
- In rows 120-126 the authors reported some references regarding global DNA methylation in patients with epilepsy after KD treatment. It would be better to report the tissue /cells from which the analyses were performed.
At lines 120-121 we specified that experiments were performed in the hippocampus.
- Similarly, in rows 127-131 please specify the genes responding in brain adenosine levels.
The genes modulated by adenosine have been added at lines 131-132
- In row 138 please report exactly what the reference 62 says. In particular the DNMTs have been studied at expression level, so, the DNMT 1, 3a, 3b are genes and must be reported in Italics and uppercase.
The text has been changed accordingly to the referee’s suggestion with gene name in italics.
- Please, similarly do for genes reported in rows 142-144.
The text has been changed accordingly to the referee’s suggestion with gene name in italics.
- The sentence in rows 145-148 is confuse and contradictory. Moreover, the relationship with cell differentiation it seems inappropriate. Please reformulate the sentence or eliminate it.
We agreed with the referee’s suggestion and we have removed the confusing sentence
- In the first part of section 4 (MicroRNAs) it would be necessary to report the preliminary results of Caradonna et al (Science and healthy meals….. Nutrients, 2020) regarding circulating miRNAs in comparison with specific food interventions which gave rise to the question about relationship between aging, age-related inflammation, circulating miRNAs, and diet.
As suggested, we added few paragraphs discussing the suggested reference (lines 235-241)
Minor revision
- Please make a double check of the references numbers. In all manuscript have been often included reference ranges with a couple of number in brackets not sequential each other.
Thank you, we found a duplicate reference that now has been removed. Other references have been checked.
Reviewer 2 Report
Dear Authors,
Your review relates to very actual topic including the elucidation of the epigenetic mechanism of action of modern ketogenic diets. The topic is novelty and prospective due to its impact on the human health.
You collected very interesting data about positive effect of KD on the obesity and several diseases. In my opinion, your review misses the comparison between positive and possibly negative effects of KD realized by epigenetic mechanisms. The input of such data will give additional value for your review.
I also would recommend to add in the table 1 to column "Possible mechanisms" - the subject of investigation - animal species , cell culture or clinical studies on human.
Author Response
We are really thank to the referee for his suggestions.
All point have been addressed as indicated below:
- In my opinion, your review misses the comparison between positive and possibly negative effects of KD realized by epigenetic mechanisms. The input of such data will give additional value for your review.
We agree with the referee. This point is very interesting. Unfortunately, no data have been published focusing on the study of epigenetic mechanism associated with adverse effects of KD. We have added this comment in the conclusion remarks (lines 273-284).
- I also would recommend to add in the table 1 to column "Possible mechanisms" - the subject of investigation - animal species, cell culture or clinical studies on human.
Thank you for this suggestion. Table 1 has now changed accordingly.
Round 2
Reviewer 1 Report
All the suggestions were addressed. So the manuscript, in my opinion, is now publishable.